# Ecological similarities and dissimilarities between donor and recipient regions shape global plant naturalizations

Shu-ya Fan [1,2], Trevor S. Fristoe[3], Shao-peng Li [1] ✉, Patrick Weigelt [4], Holger Kreft [5,6,7], Wayne Dawson [8], Marten Winter [9], Petr Pyšek [10,11], Jan Pergl [10], Franz Essl[12], Amy J. S. Davis [2] & Mark van Kleunen [2,13,14]

A central question in ecology is why alien species naturalize successfully in some regions but not in others. While some hypotheses suggest aliens are more likely to naturalize in environments similar to donor regions, others suggest they thrive in regions where certain characteristics are different. Using the native (i.e., donor) and recipient distributions of 11,604 naturalized alien plant species across 650 regions globally, we assess whether plants are more likely to naturalize in regions that are ecologically similar or dissimilar to their donor regions. Our results show that species are more likely to naturalize in recipient regions where climates are similar and native floras are phylogenetically similar to those of their donor regions, indicating that pre-adaptation to familiar biotic and abiotic conditions facilitates naturalization. However, naturalization is also more likely in regions with lower native flora diversity and more intense human modification than in the species' native range. Among all predictors, climate similarity and difference in native flora diversity emerge as the strongest predictors of naturalization success. In conclusion, ecological similarity in some factors but dissimilarity in others between donor and recipient regions promote the naturalization of alien plants and contribute to their uneven global distribution patterns.

The numbers of species that have been introduced from their native ranges (i.e., donor regions) into non-native regions (i.e., recipient regions), where they may establish self-sustaining populations (i.e., become naturalized), are steadily increasing worldwide[1,2]. Although nearly all regions of the world harbour naturalized alien plants, the geographical distribution of these species is highly uneven, with some regions acting as donors or recipients of many more species than others[1,3]. A main question is thus why certain plant species successfully naturalize in some regions but fail in others. One explanation could be the ecological similarity of the donor and potential recipient regions. For example, the climate matching hypothesis, rooted in the idea of pre-adaptation, predicts that climatic similarities between the donor and recipient regions promote naturalization[4–8]. Nevertheless, climate

is just one among many components of the environment that may influence invasion success, and the roles of other aspects of ecological similarity require further investigation[9–12].

In addition to climate, a lack of pre-adaptation to the biotic conditions encountered in recipient regions may further constrain plant naturalization[13–15]. In their native distributions, species share concurrent evolutionary histories with co-occurring species, resulting in adaptations to the same common ecological context[13,16]. Regions that are occupied by species that share an evolutionary history with those found in alien species' native ranges may therefore indicate ecological circumstances that promote species naturalization[14,17]. In recent centuries, evolutionary trajectories of species have increasingly been shaped by anthropogenic forces, and it has been argued that species

adapted to human-modified habitats in their donor regions can exploit similar opportunities provided by human activities in recipient regions[18,19]. Thus, species may be more likely to successfully naturalize in regions that have biological and anthropogenic characteristics similar to their native ranges.

While hypotheses based on pre-adaptation pose that greater ecological similarity between donor and recipient regions promotes naturalization, several other hypotheses suggest instead that greater dissimilarity between regions may promote naturalization. For example, the enemy release hypothesis describes how the absence of specific biological antagonists (i.e., competitors, pathogens, herbivores) in recipient regions can provide greater opportunities for naturalization[20,21]. Therefore, in regions with greater biotic dissimilarity to the native range of the alien species, the likelihood of shared natural enemies is expected to decrease, potentially resulting in higher naturalization success. It has also been suggested that the uneven numbers of naturalized alien plants among regions reflect differences in the invasion potential of species among regional floras[3]. The evolutionary imbalance hypothesis, for example, predicts that alien species originating from regions with higher biodiversity exhibit a greater potential to invade regions with low diversity because species from high-diversity regions may have evolved a higher competitive ability[3,22,23]. Moreover, as naturalized species are often found in habitats heavily modified by humans, higher levels of human activity in recipient regions may provide more opportunities for naturalization[24,25]. However, whether similar or greater levels of human modification are more critical for global plant naturalization remains an open question.

The relationship between ecological similarity and naturalization success likely depends on specific dimensions of ecological similarity, with alien species most likely to naturalize in regions that are similar to their native ranges in some characteristics but different in others. However, a key challenge in assessing the influence of multi-dimensional ecological similarity lies in the nature of ecological differences, which can be either unidirectional or bidirectional. Unidirectional dimensions, such as geographical distance and dissimilarity in floristic composition, represent absolute differences between regions. In contrast, bidirectional dimensions, such as climate variables and native flora diversity, can take on positive or negative values, reflecting whether a recipient region is, for example, warmer or colder, or has a higher or lower diversity than the native region. In either case, the relationship between ecological distances and naturalization success can be analyzed within a unified framework based on generalized linear models with linear and quadratic terms (detailed descriptions are provided in Box 1). This framework allows for the identification of scenarios where species have the highest likelihood of naturalization in regions that are dissimilar, moderately dissimilar, or similar to their native range across multiple ecological dimensions (Box 1).

Here, we apply our analytical framework to test how ecological distances between potential non-native recipient regions and native donor regions of alien species relate to their observed naturalization patterns. We first compile a dataset comprising the donor and potential recipient regions ($n = 650$) of 11,604 naturalized angiosperm plant species worldwide (Supplementary Figs. 1 and 2). For each species, we calculate distances for different ecological factors between their donor and potential recipient regions, including bidirectional distances for temperature, precipitation, human modification, and native flora diversity, as well as the unidirectional distance for phylogenetic dissimilarity of the native floras (Table 1 and Supplementary Fig. 3). Furthermore, given that species might have a higher likelihood of introduction to regions near their native distribution, and because spatially adjacent regions typically exhibit similar environmental conditions[26,27], we also quantify the geographical distance between recipient and donor regions, which is also a unidirectional distance. We then use multivariate generalized linear models with both linear and

quadratic terms (see Box 1) to quantify the shape of the relationship between each distance and naturalization probability, and further evaluate their relative contributions to explaining global patterns of naturalization. We show that ecological distances across both unidirectional and bidirectional dimensions jointly shape naturalization success, with climatic similarity and dissimilarity of native-flora diversity identified as the strongest factors associated with naturalization success. Together, these findings advance our understanding of how multiple ecological dimensions shape the uneven global distribution of naturalized plants.

## Results
### Effects of ecological distances on plant naturalization
Using a multivariate generalized linear mixed-effects model (GLMM), we found that similarities in different ecological dimensions showed varying effects on plant naturalization probability in potential recipient regions (Fig. 1). We assessed the bidirectional climatic distances between donor and recipient regions using the first two axes of a principal component analysis (PCA) on 19 bioclimatic variables[28]. The first axis ($PC_{Temp}$) represented temperature and its seasonality, while the second axis ($PC_{Prec}$) reflected precipitation and its seasonality (Supplementary Table 1). When other ecological distance metrics were held constant, naturalization probability peaked when $PC_{Temp}$ distance was close to zero, indicating that species were more likely to naturalize in regions with similar temperature conditions, specifically annual mean temperature and temperature seasonality (Figs. 1a and 2a and Supplementary Table 2). Compared to $PC_{Temp}$ distance, we found that, for $PC_{Prec}$ distance, naturalization probability peaked in recipient regions that were moderately wetter and had more stable precipitation compared to the donor regions (Figs. 1b and 2a and Supplementary Table 2).

In contrast to climatic similarities, the effects of bidirectional distances in human modification and native flora diversity were strongly directional. Specifically, the directional effect of human modification distances, assessed by comparing the human modification index (HMI) of recipient and donor regions, also significantly influenced naturalization probability. Species were more likely to naturalize in recipient regions with greater human modification than in their native range, though the probability declined slightly at the highest levels of human modification differences (linear term: $\beta_1 = 0.39$, $z = 12.79$, $P < 0.001$, 95% CI [0.33, 0.45]; quadratic term: $\beta_2 = -0.09$, $z = -19.70$, $P < 0.001$, 95% CI [−0.10, −0.08]; Fig. 1c and Supplementary Table 2). Moreover, bidirectional differences in native flora phylogenetic diversity between recipient and donor regions exhibited a significantly negative relationship with naturalization probability (linear term: $\beta_1 = -1.82$, $z = -48.46$, $P < 0.001$, 95% CI [−1.90, −1.75]; quadratic term: $\beta_2 = -0.03$, $z = -4.52$, $P < 0.001$, 95% CI [−0.04, −0.02]; Figs. 1d and 2a and Supplementary Table 2). This result was consistent when measuring distance based on taxonomic rather than phylogenetic flora diversity (Fig. 1d and Supplementary Fig. 4).

Beyond these bidirectional differences, unidirectional ecological distances, including floristic dissimilarity and geographical distance, also contributed to variation in naturalization probability. After accounting for the effects of other ecological distance metrics, the relationship between floristic dissimilarity and species naturalization probability showed that the likelihood of naturalization significantly increased when recipient regions had native floras that were more similar to the native ranges (linear term: $\beta_1 = -0.45$, $z = -16.28$, $P < 0.001$, 95% CI [−0.50, −0.40]; quadratic term: $\beta_2 = -0.07$, $z = -16.34$, $P < 0.001$, 95% CI [−0.08, −0.06]). This was true irrespective of whether we used Simpson's phylogenetic dissimilarity (i.e., the degree of shared evolutionary history between regions; Fig. 1e and Supplementary Table 2) or Simpson's taxonomic dissimilarity (i.e., the degree to which species are shared between regions; Supplementary Fig. 5). Furthermore, we found that the probability of successful

# BOX 1
# Analytic framework

The relationship between ecological distances (unidirectional and bidirectional distances) and naturalization probability can be analyzed using generalized linear models ($\text{logit}(y) = \alpha + \beta_1 x + \beta_2 x^2$), that predict naturalization probability through linear ($\beta_1$) and quadratic ($\beta_2$) terms for a given ecological distance metric.

When conditions similar to those in the donor regions are most favorable for naturalization, the probability peaks at an ecological distance of zero. For bidirectional differences, this is represented as $\beta_2 < 0$ and $\beta_1 = 0$ in the model (**c**). For unidirectional differences, it is expressed as $\beta_2 = 0$ and $\beta_1 < 0$, indicating that naturalization probability is highest under conditions closest to those in the donor regions (**h**).

When naturalization is favored in recipient regions that differ from donor regions, bidirectional differences may only promote naturalization in specific directions (**a, e**). For instance, the evolutionary imbalance hypothesis predicts that species introduced from

biodiverse donor regions to low-diversity recipient regions are more likely to naturalize. In such cases, $\beta_2 = 0$, while significant linear terms indicate whether naturalization probability is highest under conditions substantially higher ($\beta_1 > 0$; **a**) or lower ($\beta_1 < 0$; **e**) than donor regions. For unidirectional differences, any form of difference may be beneficial for naturalization ($\beta_2 = 0$ and $\beta_1 > 0$, **f, j**).

Naturalization success may also be highest when conditions are different, but not too different from those in the donor regions. In other words, there may be an optimal difference that facilitates naturalization ($\beta_2 < 0$ in the model). For bidirectional differences, linear terms indicate whether the highest naturalization probability occurs under moderately higher ($\beta_1 > 0$; **b**) or moderately lower ($\beta_1 < 0$; **d**) environmental conditions than in donor regions. For unidirectional differences, any moderate difference may promote naturalization ($\beta_1 > 0$; **g, i**).

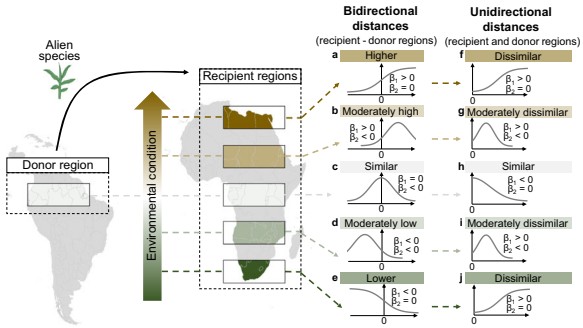

**A schematic diagram illustrating potential relationships of ecological distances between donor and recipient regions with the naturalization probability of alien species.** The left panel shows the gradient of ecological conditions in recipient regions relative to donor regions, which alien species may encounter. The right panel shows the expected relationships between ecological distances and naturalization probability under scenarios where the ecological conditions in a given recipient region favours naturalization. The scenarios **a**–**e** show the relationships for bidirectional distances, where distances can be positive (gold) or negative (green), representing recipient regions with higher or lower environmental values compared to donor regions, respectively. The scenarios **f**–**j** show the relationships for unidirectional distances, where distances are always positive, with larger distances (darker colors, either gold or green) indicating greater ecological dissimilarity between recipient and donor regions. $\beta_1$ and $\beta_2$ represent the coefficients of linear and quadratic terms of the generalized linear mixed model, respectively, used to describe the relationship between ecological distances and naturalization probability. Maps were generated using polygons from the GloNAF database.

naturalization was highest at intermediate geographical distances between donor and recipient regions (c. 15,000 km), showing a significant nonlinear association (linear term: $\beta_1 = 0.95$, $z = 43.10$, $P < 0.001$, 95% CI [0.91, 1.00]; quadratic term: $\beta_2 = -0.12$, $z = -29.70$, $P < 0.001$, 95% CI [−0.13, −0.11]; Fig. 1f and Supplementary Table 2).

### Relative contributions of ecological distances
By assessing the relative importance of each ecological distance variable (combining the linear and quadratic terms), we found that $PC_{Temp}$ distance was the most important predictor of global plant naturalization probability. Averaged across 999 resampled models (see "Methods"), $PC_{Temp}$ distance accounted for 56.21% of the explained fixed-effect variance (95% CI [55.63%, 56.79%]; Fig. 2b and Supplementary Table 3). This contribution was mainly attributed to the strong effect of its quadratic relationship with naturalization probability (quadratic term: $\beta_2 = -1.32$, $z = -152.21$, $P < 0.001$, 95% CI [−1.34, −1.30]; Supplementary Table 2), indicating that temperature similarity, rather than the direction toward warmer or cooler climates, is a key determinant of global plant naturalization success. $PC_{Prec}$ distance also made a considerable contribution to the model's variance (average 13.53%, 95% CI

[13.15%, 13.92%]; Fig. 2b and Supplementary Table 3), with both its linear and quadratic components having significant effects (linear term: $\beta_1 = 0.53$, $z = 20.22$, $P < 0.001$, 95% CI [0.48, 0.58]; quadratic term: $\beta_2 = -0.20$, $z = -47.86$, $P < 0.001$, 95% CI [−0.21, −0.19]; Supplementary Table 2), suggesting that naturalization success was highest when recipient regions were moderately wetter than donor regions, while larger differences in precipitation reduced the likelihood of establishment. Another important predictor was the bidirectional difference in native flora phylogenetic diversity, which accounted for 20.21% of the explained variance on average (95% CI [19.75%, 20.66%]; Fig. 2b and Supplementary Table 3), and showed a strong negative linear association with naturalization probability (Supplementary Table 2), while the quadratic term had a negligible effect (Supplementary Table 2). In contrast, distances in human modification, floristic phylogenetic dissimilarity, and geographical distance, while statistically significant, each explained only a marginal share of the variance.

### Consistency of the results across alternative scenarios
To assess the robustness of our findings, we additionally conducted sensitivity analysis using different sets of potential recipient regions

## Table 1 | Quantification of ecological and geographical distance metrics between donor and recipient regions

| | Variables | Definition | Explanation |
|---|---|---|---|
| **Bidirectional distance** | PC$_{Temp}$ distance | Area-weighted mean difference in average PC$_{Temp}$ values across all grid cells within the recipient and donor regions (recipient region minus donor regions) | A positive value indicates that the recipient region is warmer with more stable temperature than the donor regions, while a negative value means it is colder with more seasonal temperatures. |
| | PC$_{Prec}$ distance | Area-weighted mean difference in average PC$_{Prec}$ values across all grid cells within the recipient and donor regions (recipient region minus donor regions) | A positive value indicates that the recipient region is wetter with less seasonal rainfall than the donor regions, while a negative value means it is drier with seasonal rainfall. |
| | Human modification distance | Area-weighted mean difference in average HMI values across all grid cells within the recipient and donor regions (recipient region minus donor regions) | A positive value indicates that the recipient region has a higher degree of human modification than the donor regions, while a negative value indicates a lower degree. |
| | Native flora phylogenetic diversity distance | Difference in the area-corrected phylogenetic diversity of the native flora between the recipient and donor regions (recipient region minus donor regions) | A positive value means that the recipient region's native flora has a higher phylogenetic diversity than the donor regions, while a negative value means it is lower. |
| **Unidirectional distance** | Floristic phylogenetic dissimilarity | Area-weighted mean of pairwise phylogenetic Simpson's dissimilarity indices between the recipient and donor regions | A larger value indicates less shared evolutionary branch length between the floras of recipient and donor regions, indicating they are more phylogenetically dissimilar in their floristic composition. |
| | Geographical distance | Area-weighted mean of the pairwise distances between the geographical centroids of the recipient and donor regions | A larger value indicates that the recipient region is geographically more distant from the donor regions. |

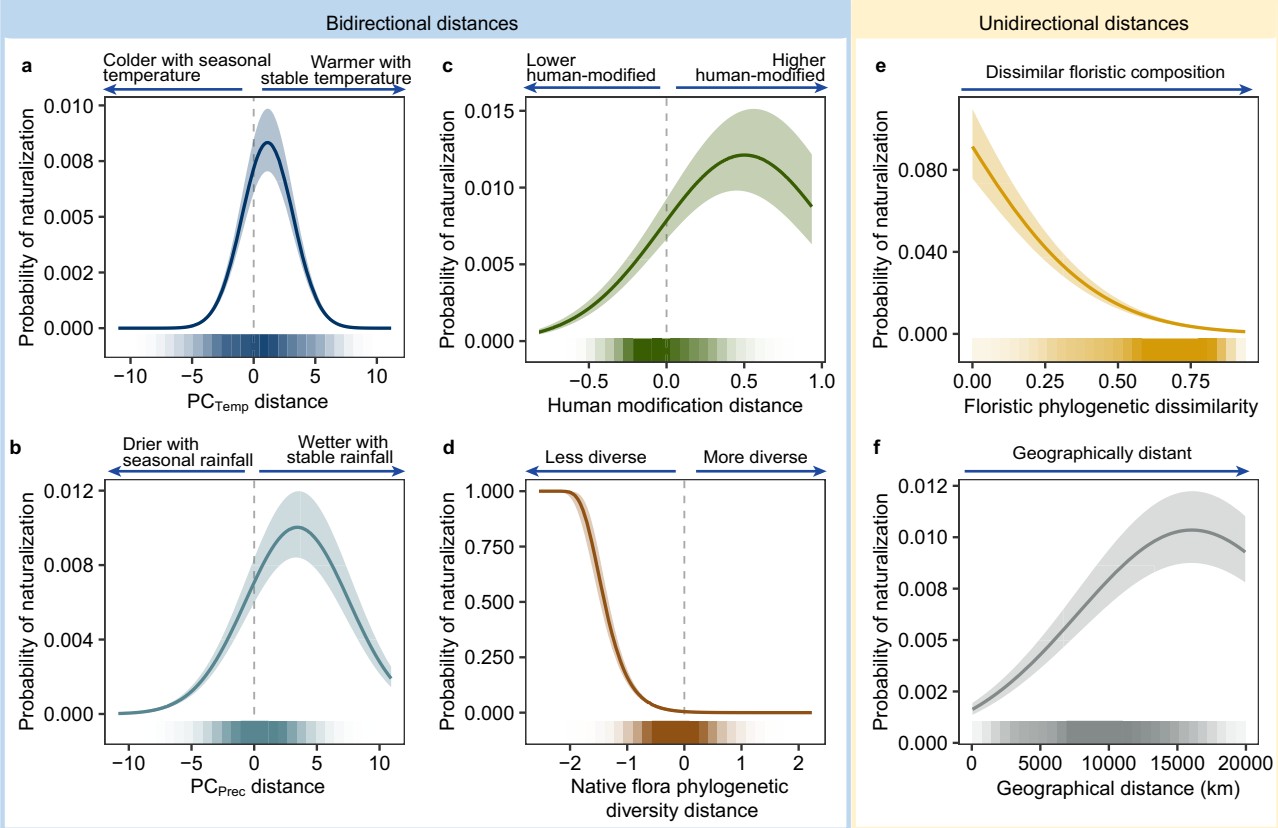

**Fig. 1 | Partial relationships between naturalization probability and ecological distances between recipient and donor regions.** The multivariate generalized linear mixed-effects model, based on 6,931,789 observations of 11,604 alien plant species naturalized across 650 regions, included linear and quadratic terms for each of the six ecological distances (**a**–**f**). Positive values of the bidirectional distance indicators indicate that recipient regions have higher PC$_{Temp}$, PC$_{Prec}$, human modification, and native flora phylogenetic diversity than the donor regions, while negative values indicate the opposite. Larger values for unidirectional distances indicate greater dissimilarity in phylogenetic floristic composition, as well as increased geographical distance between the recipient and donor regions. The solid lines represent the predicted mean (model fit), and the shaded areas denote the 95% confidence intervals of these predictions. The standardized coefficients for the linear and quadratic terms are shown in Fig. 2. The bar below each plot indicates the number of data points in each bin, where the predictor variable was divided into 30 segments, with darker shades indicating more data points within each bin. Note that the *y*-axis scales differ across panels to reflect variations in data range. Source data are provided as a Source Data file.

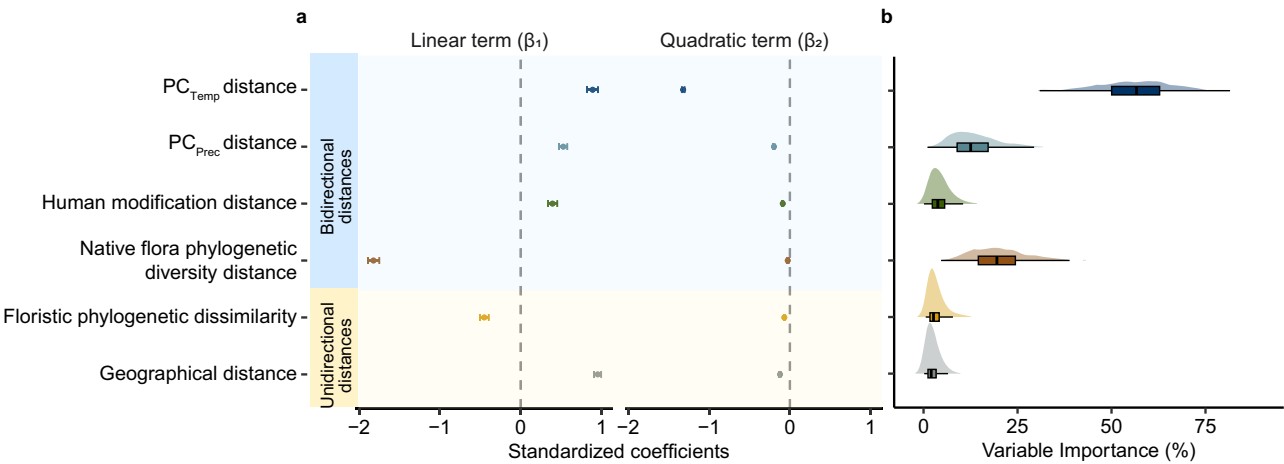

**Fig. 2 | Standardized coefficients and relative importance of the ecological distances between recipient and donor regions on naturalization probability.** **a** Standardized coefficients of the linear ($\beta_1$) and quadratic ($\beta_2$) terms of ecological distances, estimated using a multivariate generalized linear mixed-effects model based on 6,931,789 observations of 11,604 alien plant species naturalized across 650 regions. Points represent the standardized coefficients for the linear and quadratic terms of each distance metric, and error bars indicate their 95% confidence intervals (from model-based standard errors); in some cases, the intervals are narrower than the symbols and thus not visible. All standardized coefficients are significantly different from zero (all *P* values from two-sided Wald *z*-tests <0.001).

No adjustments for multiple comparisons were applied. **b** Distributions of variable relative importance for each ecological distance, calculated as the combined contributions of their linear and quadratic terms across 999 bootstrap replicates, with each replicate fitted on 5000 randomly selected observations from the complete dataset of 6,931,789 observations. Density plots (top) show the full distribution of relative importance values (%), and boxplots (bottom) show the median (center line), the upper and lower quartiles (box limits), and the largest and smallest values within 1.5× the interquartile range (whiskers); outliers are not shown. Source data are provided as a Source Data file.

and subsets of naturalized alien plants. These included restricting potential recipient regions to continents where species were already naturalized, limiting the number of potential recipient regions, and analyzing subsets of species with economic uses or widespread distributions (see "Methods" for details). Across all scenarios, similar temperature conditions, lower native flora phylogenetic diversity, and higher human modification in recipient regions consistently facilitated naturalization success (Supplementary Fig. 6). As some of the potential recipient regions in our dataset are islands, we additionally tested whether results differed between island and mainland regions. We found that main patterns were similar between the two groups of regions (Supplementary Fig. 7).

## Discussion

We assessed how different components of ecological similarity between the donor and potential recipient regions related to naturalization probability for 11,604 naturalized alien plants globally. Our results revealed that similarity in climate and differences in native plant diversity between donor and recipient regions are the most important factors shaping global patterns of plant naturalization. Alien plants were more likely to naturalize in regions with temperature conditions similar to their donor regions, while differences in other environmental factors, such as moderately wetter and lower native flora diversity in recipient regions, also facilitated naturalization success. Our findings underscore the importance of assessing multidimensional ecological similarities to reveal how specific ecological distances between donor and recipient regions influence the naturalization success of alien species globally.

Plant species were more likely to naturalize in regions that have similar temperature conditions to their donor regions, highlighting the importance of climatic matching in facilitating naturalization success. Among all ecological dissimilarities considered, temperature similarity showed the strongest association with naturalization probability, surpassing other biological or anthropogenic similarity metrics. This pattern further supports the long-standing view that climatic pre-adaptation, particularly to temperature, is a crucial factor influencing species naturalization[7,8,29]. In addition to climatic

pre-adaptation, human preferences for introducing certain plants also likely contribute to these patterns. For example, in Germany, horticultural introductions often target species with traits like frost tolerance, which aligns closely with the local climate[30]. Similarly, in southern Africa, many species introduced for cultivation originated from regions with comparable climatic conditions[31]. Thus, both ecological filtering through pre-adaptation and human introduction preferences for climate matching likely contributed to the increased likelihood of naturalization in climatically similar regions[30].

Although our results suggest the importance of climatic suitability, naturalization probability peaked in regions with temperature conditions that were slightly warmer and less seasonal than those of the species' native ranges. Additionally, the probability of naturalization peaked when the recipient region was moderately wetter and more stable than the native region. One possible explanation is that in their native ranges, species may not fully occupy their fundamental climatic niches due to biotic interactions. This may be particularly the case under stable, warm, and wet climates, where biotic interactions are often intense (species interaction-stress hypothesis[32]). In recipient regions, however, the alleviation of non-climatic barriers such as natural enemies may allow species to occupy parts of their fundamental climatic niches, such as wetter environments, from which they are excluded in their native ranges[33]. The wetter conditions may offer additional opportunities that facilitate the colonization and spread of naturalized species. Other studies have also shown that many global invaders originating from drylands tend to establish in wetter environments, including forests and riparian zones[9]. Similar patterns have been observed in other taxonomic groups. A global analysis of birds found that naturalized species on islands, where native diversity and ecological resistance are generally lower, are often able to occupy latitudes lower than those in their native ranges[34]. These low-latitude regions are usually characterized by warmer and wetter climates. While our data do not allow us to explicitly test mechanisms such as biotic interactions or niche shift, the observed patterns seem to be consistent with the idea that biotic release can open new climatic opportunities for naturalization of alien species[20,21].

In addition to the importance of climatic similarity, we found that the floristic similarity between the native floras in recipient regions and those in donor regions was positively related to naturalization success of alien species. The composition of the native flora serves as a comprehensive indicator of environmental suitability, reflecting long-term adaptation of floras to specific biotic and abiotic conditions, including those beyond the macroclimatic variables included in our analyses[16]. A higher taxonomic and phylogenetic similarity of the native floras suggests that recipient regions share more native species and evolutionary history with the donor regions. This could be associated with similar abiotic environmental filters, and potentially indicates the presence of suitable mutualistic partners in the recipient regions, such as pollinators and seed dispersers[16,17,35]. However, this assumption requires further confirmation through direct comparisons and field observations of communities of pollinators and dispersers, as well as their traits, between recipient and donor regions. Although floristic similarity had a positive effect, its contribution to naturalization success was relatively weak compared to climatic similarity at the global scale. This contrasts with findings from regional-specific studies, such as those in eastern North America, where floristic similarity was identified as a stronger predictor of naturalization success than climatic similarity[17]. Such differences may reflect variation in the invasion stages considered and the spatial extent analyzed[36,37]. Climatic similarity usually acts as a primary constraint during the naturalization stage and may be further strengthened when considering greater environmental heterogeneity at global scales, whereas during the spread stage, its constraining effect may weaken[38]. Therefore, for predictions of distributions of alien species across invasion stages, it is crucial to consider not only climatic matching between native regions and regions of introduction but also similarity of the native floristic composition and evolutionary history[16,17]. This consideration would be particularly crucial for high-latitude regions, where harsh environmental conditions may intensify the role of pre-adaptation in species naturalization[39].

Although we found that environments similar to the ones in donor regions significantly facilitate naturalization, we also observed that moderate differences in certain ecological variables can further facilitate species naturalization. Consistent with the predictions of the intermediate-distance hypothesis[40,41], we observed that species had the highest naturalization probability at intermediate geographical distances from their donor regions. This hypothesis explains the pattern as arising from a trade-off between colonization pressure, which increases with distance because species from more distant donor regions are more likely to be absent from (i.e., not native to) the recipient region, and post-introduction survival, which decreases as environmental differences increase[40,41]. Interestingly, our analyses recover an intermediate-distance pattern even while considering only recipient regions outside the species' native distributions and accounting for key climatic factors, suggesting alternative mechanisms from those originally proposed to explain the hypothesis. The results may reflect the influence of other unmeasured variables, such as abiotic factors (e.g., soil conditions or land use) and distance-related drivers, including trade flows and historical dispersal processes, which should be considered in future studies. A previous study found that for plants, the optimal distance for biological invasion was c. 10,000 km[41], whereas we found it to be c. 15,000 km. This discrepancy could reflect differences in the methods used to calculate distances and the fact that the previous study was limited to c. 800 invasive plant species[41], while our dataset included over 11,000 naturalized alien plant species. Nevertheless, both studies are consistent with patterns of intermediate introduction distances favouring naturalization.

Differences in human activity and plant diversity between recipient and donor regions showed significant and directional influences on species naturalization. Specifically, alien species are more likely to naturalize in regions that have been more intensively modified by

humans than their donor regions, rather than in regions with similar levels of modification. This may be because human-modified environments often offer additional resources and ecological opportunities, such as increased nutrient availability and modified microhabitats, which may suppress native species and facilitate the establishment of alien species[24,25,42,43]. Moreover, regions with intense human activity typically coincide with dense human populations, intense trade, and extensive transportation networks, which together increase opportunities for alien species introductions, cultivation, and spread[43–45]. For example, Europe imports about 19% more goods from Asia than it exports to Asia, potentially increasing the introduction pressure from Asia to Europe[46]. Such asymmetries in global trade flows may further amplify the probability of species introductions into highly disturbed regions[45]. However, beyond a certain threshold, recipient regions with substantially higher levels of human modification compared to donor regions exhibited reduced naturalization probability. This may be because excessive human modification—such as areas dominated by impervious surfaces or dense urban infrastructure—limits the availability of natural spaces for plant population establishment. In general, our results showed that differences in human activity between donor and recipient regions were a weaker predictor of naturalization success than temperature similarity, despite previous studies highlighting anthropogenic factors as strong drivers of naturalization[44,47,48]. This suggests that the local intensity of human activity in recipient regions, rather than donor-recipient differences, may be a more important factor driving naturalization. In other words, even if donor and recipient regions differ little in overall human activity levels, recipient regions with high levels of human disturbance may still offer opportunities for naturalization. It is also possible that the composite human modification index used here did not fully capture the specific dimensions of human influence most relevant to naturalization outcomes. Future research could benefit from integrating both the local intensity of human activity and its dissimilarity to donor regions, alongside more specific indicators, such as trade volume, transport connectivity, and propagule and colonization pressures, to clarify how human influence shapes global patterns of plant naturalization.

We also found that species were more likely to naturalize in recipient regions that had a lower native phylogenetic or taxonomic diversity than the donor regions. This finding aligns with the predictions of the evolutionary imbalance hypothesis, which posits that species from regions with high biodiversity have undergone prolonged intense competition, resulting in a higher competitive ability that facilitates naturalization[22,23]. A recent global analysis also supports this idea, showing that species from biodiverse regions are more likely to establish naturalized populations outside their native ranges[3]. Although differences in native plant diversity between regions have received limited attention in global studies, our model identified it as the second most important predictor of global naturalization probability, following temperature similarity. Similar patterns have been observed at regional scales; for example, in New Zealand forests, the phylogenetic diversity of the donor region was identified as the strongest predictor of community invasibility[22]. It is possible that the global pattern identified by our study partly reflects the fact that many species from mainland regions, usually characterized by high biodiversity, have become naturalized on islands, which usually have a low biodiversity and are highly susceptible to invasions[49,50]. However, our analyses indicate that this pattern holds consistently across both island and mainland regions (Supplementary Fig. 7), suggesting its robustness across different geographical contexts, rather than being solely attributable to island effects.

Although additional sensitivity analyses confirmed the robustness of our results, we acknowledge limitations of our study that leave opportunities for future advancements when more detailed datasets become available. One limitation is the lack of geographically explicit

data on species introductions at the spatial resolution of our analysis, which limited our ability to account for actual naturalization failures of introduced aliens. As spatially explicit records of alien species introductions become more widely available[51,52], they would help address this gap and provide better understanding of the factors that determine species naturalization success. Furthermore, our analysis was limited by the spatial resolution of available floristic inventory data, which typically is at the level of administrative regions such as countries or provinces. This broader scale may overlook finer-scale habitat differences and ecological interactions (e.g., competition and mutualisms), which are often crucial at the community level[53]. Future studies could benefit from higher-resolution data to capture these habitat differences and ecological interactions and investigate how they influence naturalization success.

As the numbers of naturalized alien species continue to rise[2], it becomes increasingly crucial to understand the global patterns and mechanisms of species naturalization. By assessing the relationship between naturalization probabilities and ecological distances between donor and recipient regions for naturalized alien plants around the world, we showed that whether species prefer recipient regions that are similar or dissimilar to their donor regions depends on the ecological factor considered. Our results thus underscore the necessity of considering multidimensional ecological dissimilarities and their directional aspects, to understand global patterns of species naturalization, and improve predictions of regional invasion risk. Notably, under increasing human disturbance[54], climate change[55], and trade[45,56], future efforts should focus on the potential spread of species from high-diversity regions within the same biogeographic zone to low-diversity regions and rapidly urbanizing areas, which are likely to emerge as hotspots for global plant naturalization and invasion.

## Methods

### Donor and recipient distributions of naturalized alien plants

To obtain the donor and recipient distributions of naturalized alien plants around the globe, we first extracted the list of naturalized alien plant species and the regions in which each of those species is naturalized from version 2.0 of the Global Naturalized Alien Flora (GloNAF) database[1,57]. The GloNAF database includes ~16,000 naturalized alien vascular plant species and covers over 1300 regions (e.g., countries, states, provinces, islands)[1,57]. The native distributions (i.e., the donor regions) of these naturalized alien plants were retrieved from version 3.0 of the Global Inventory of Floras and Traits (GIFT) database using the R package GIFT v.1.2.0[58]. This database contains the native distributions of approximately 370,000 plant species across nearly 3400 regions worldwide[58,59]. We selected regions present in both GloNAF and GIFT, and we also merged certain regions from the GloNAF database to match larger regions in GIFT, and vice versa.

Because regional checklists for non-angiosperms are usually not as comprehensive as those for angiosperms (i.e., flowering plants), our study focused on naturalized angiosperms only. As both GloNAF and GIFT use the World Checklist of Vascular Plants (WCVP)[60,61] as taxonomic reference list, species names in both databases could be matched directly. We only retained species names that are accepted in WCVP. Hybrids were excluded from our analysis, and infraspecific taxa (such as varieties and subspecies) were assigned to the binomial species level. Furthermore, if GloNAF identifies a species as naturalized in a particular region, but GIFT indicates it is native to that region, we categorized the species as native and removed the naturalization record. We note that our main results remain consistent even if these species with conflicting status were classified as naturalized (Supplementary Fig. 8). We excluded naturalized alien plants for which native distribution data were not recorded in our dataset. This might be species that are only known from cultivation.

Consequently, our dataset includes the native and naturalized distributions of 11,604 naturalized angiosperm species across 650 regions globally, covering approximately 85% of the global ice-free land surface.

### Quantification of ecological distances between recipient and donor regions

To compare the ecological distances between donor and potential recipient regions of naturalized alien plants, we defined the latter as all regions outside the species native distribution. These potential recipient regions encompass regions where the species has successfully naturalized as well as those where it has not. For each naturalized alien plant, we measured the distances between each of its donor and potential recipient regions as the bidirectional distances with regard to climate, diversity of the native floras, the degree of human-induced landscape modification, as well as the unidirectional distances for composition of the native floras, and geographical location.

To quantify climatic distances between the donor and potential recipient regions, we obtained all 19 bioclimatic variables at 10 arc-minutes resolution from WorldClim version 2.1[28]. Given the collinearity among the bioclimatic variables, we conducted a PCA. Before the PCA, certain bioclimatic variables, due to their skewed distributions, were transformed to obtain approximately normally distributed variables by using the R package normalizer v.0.1.0 (Supplementary Table 1)[62]. All transformed variables were scaled to a mean of zero and a standard deviation of one. The PCA was then conducted based on these standardized bioclimatic variables. The first two principal component axes ($PC_{Temp}$ and $PC_{Prec}$) explained 44.42% and 33.25% of the climatic variation, respectively, accounting for a total of 77.67% (Supplementary Table 1). Higher $PC_{Temp}$ and $PC_{Prec}$ values indicated warmer, less seasonal temperatures and wetter, less seasonal precipitation, while lower values corresponded to colder, drier conditions with stronger seasonality (Supplementary Table 1). For each potential recipient region of a naturalized alien plant, the climatic distances to the donor regions were calculated as the area-weighted mean of pairwise differences in average $PC_{Temp}$ and $PC_{Prec}$ values. Specifically, we first computed the average $PC_{Temp}$ and $PC_{Prec}$ values for all grid cells within each region, including those fully or partially overlapping with the region. To account for variation in the sizes of donor regions, we used the area of each donor region as a weight in the calculation of the area-weighted mean for each recipient-donor region pair. Positive values of bidirectional distance in $PC_{Temp}$ and $PC_{Prec}$ indicated that the average climate in the recipient region was warmer, wetter, and less seasonal than in the species' donor regions. Conversely, negative values indicated that the recipient region was colder, drier, and more seasonal than in the species' donor regions.

To quantify the distance in the degree of human-caused landscape modification, we used the human modification index (HMI) at $1 km^2$ resolution, which comprehensively assesses the extent of modification of terrestrial lands due to 13 anthropogenic stressors, including variables related to human settlement, agriculture, transportation, mining and energy production and electrical infrastructure[63]. HMI ranges from 0 to 1, with increasing values corresponding to progressively higher degrees of human modification of the landscape[63]. For each potential recipient region of a naturalized alien plant, the bidirectional distance in HMI was defined as the area-weighted mean of pairwise differences in the mean HMI values between the potential recipient region and each donor region. The mean HMI values for each region were derived from the average of all grid cells overlapping the region, and the pairwise differences were weighted by the area of the donor regions. Positive values indicated that the recipient region exhibited a higher level of human modification than the donor regions, while negative values

indicated a lower level of human modification in the recipient region compared to the donor regions.

To calculate the bidirectional distance in diversity of native floras between potential recipient and donor regions, we assessed differences in taxonomic and phylogenetic diversity of native floras. Taxonomic diversity was measured as the number of native species (i.e., species richness), and phylogenetic diversity was calculated as Faith's phylogenetic diversity (i.e., Faith's PD)[64] using the R package picante v.1.8.2[65]. The native angiosperm flora for each of the 650 regions was obtained from the GIFT database[58]. For calculating Faith's PD, we used the comprehensive mega-phylogeny provided by the R package GIFT v.1.2.0[58], encompassing approximately 350,000 angiosperm species. For species absent from the mega-phylogeny, we incorporated them into the phylogenetic tree manually using the R package phytools v.2.1.1[66]. Specifically, 33 species were grafted at the roots of their respective genera. Five species whose genera were entirely missing were grafted at the roots of their respective families. We then pruned the phylogeny to include only the 287,070 species analysed in this study. Given the considerable variation in the area of regions from which naturalized alien plants originate and the strong association between the floristic diversity and regional area[67], we used area-corrected measures of taxonomic and phylogenetic diversity for our analyses. Specifically, we first calculated the area, Faith's PD, and species richness for each of the 650 regions, as well as the total area, Faith's PD, and species richness of all native species within entire native distributions of each naturalized alien plant. So, if a naturalized alien plant is native in ten regions, we calculated the cumulative area of these 10 regions as well as Faith's PD and species richness for the combined flora of these 10 regions. We then performed log-log linear regressions of Faith's PD and species richness against their respective areas, respectively. The area-corrected phylogenetic diversity and area-corrected taxonomic diversity were then derived from the residuals of these regressions (Supplementary Fig. 9). Thus, for a potential recipient region of a given naturalized alien plant, the bidirectional distances in biodiversity were calculated by comparing the area-corrected phylogenetic and taxonomic diversities of the recipient region to those of the donor region. Positive values indicate that the potential recipient region has higher phylogenetic and taxonomic diversities than the native distribution, while negative values indicate lower phylogenetic and taxonomic diversities. Given the strong correlation between area-corrected phylogenetic diversity and area-corrected taxonomic diversity (Pearson's correlation coefficient: $r = 0.98$, $P < 0.001$; Supplementary Fig. 10), we chose to use area-corrected phylogenetic diversity as our biodiversity metric, because it offers a more comprehensive measure by incorporating evolutionary history. Additionally, models incorporating area-corrected phylogenetic diversity had lower AIC values than those using area-corrected taxonomic diversity ($\Delta AIC = 787$), suggesting that the effect of phylogenetic diversity is not solely driven by taxonomic diversity. Nevertheless, results based on area-corrected taxonomic diversity were consistent with those using area-corrected phylogenetic diversity (Supplementary Fig. 4).

To quantify the unidirectional distance in floristic composition between potential recipient regions and donor regions, we calculated the taxonomic and phylogenetic dissimilarities of their native floras. Specifically, we used R package betapart v.1.5.4[68] to calculate the Simpson taxonomic dissimilarity and Simpson phylogenetic dissimilarity, which measure the proportion of shared species and shared evolutionary branch lengths between two floras, respectively[69,70]. These indices are considered robust against differences in species richness between regions[68]. Thus, for each potential recipient region of a given naturalized alien plant, we calculated the area-weighted mean of pairwise Simpson's taxonomic and phylogenetic dissimilarity indices between the potential

recipient region and each donor region. Both taxonomic and phylogenetic dissimilarity values range from 0 to 1, with greater values indicating fewer shared species and less shared evolutionary branch lengths between the floras of donor and recipient regions. However, given the strong correlation between taxonomic and phylogenetic dissimilarities (Pearson's correlation coefficient: $r = 0.72$, $P < 0.001$; Supplementary Fig. 10), we chose to present only phylogenetic dissimilarity in the main text as it may better capture the flora's long-term evolutionary adaptations to specific environments. Additionally, the model with phylogenetic dissimilarity had a lower AIC than the one with taxonomic dissimilarity ($\Delta AIC = 4069$). Nonetheless, the results based on taxonomic dissimilarity are consistent with those of phylogenetic dissimilarity presented in the main text (Supplementary Fig. 5).

For a given naturalized alien plant, the geographical distance of each potential recipient region to the donor regions was calculated as the area-weighted mean of the pairwise distances between their geographical centroids using the R package geosphere v.1.5-14[71]. We also calculated the area-weighted mean of the nearest distance between the geographical boundaries of each potential recipient region and the donor regions. As the distances between geographical centroids and regional boundaries were highly correlated (Pearson's correlation coefficient: $r = 0.996$, $P < 0.001$; Supplementary Fig. 10), the main results were consistent when using distances based on geographical nearest distance (Supplementary Fig. 11).

## Statistical analyses

We modelled the naturalization success of an alien plant in a potential recipient region as a function of the various ecological distances and geographical distance, using a GLMM. We accounted for the statistical non-independence of multiple records per region and per species by incorporating region ID and species ID as random effects. Given that naturalization success is a binary response variable (naturalization vs. non-naturalization), we used a binomial distribution for our analysis. Due to the higher number of non-naturalization cases (zeros) compared to naturalization cases (ones), we used the complementary log–log link function, which is suitable for asymmetric data[72]. All subsequent GLMM analyses used the same link function and included the same random effects. As fixed terms, we included linear and quadratic terms of the bidirectional distances of $PC_{Temp}$, $PC_{Prec}$, the degree of human modification, and the phylogenetic diversity of the native floras, as well as the phylogenetic dissimilarity of the native floras and the geographical distance. To visualize the effect of each variable on the probability of species naturalization, we plotted the predicted partial relationships between the probability of species naturalization and each distance metric while holding other variables constant at their mean values. To facilitate comparisons of the effects of the different ecological distance variables, we also calculated the standardized coefficients for each fixed effect.

The above model makes the assumption that each species had the opportunity to naturalize in each potential recipient region. However, it could be that a species failed to naturalize in some of those regions because it has never been introduced there. To account for this, we also conducted sensitivity analyses using several subsets of the data. First, because an alien species is more likely to have been introduced to regions on the same continents where it has become naturalized, for each species, we restricted the potential recipient regions to those that are on continents where the species has become naturalized in at least one region. Second, we assumed potential recipient regions to include all non-native regions globally, but restricted the set of species to those that have known economic uses, because these species are likely to have been introduced to many regions. From the World Checklist of Useful Plant Species

(WCUPS)[73], we selected naturalized alien plants with economic uses that previously were found to have strong associations with naturalization success[74]: animal foods, environmental uses, gene sources, human foods, materials, and medicines. The names of the species in WCUPS were standardized according to WCVP[60,61], resulting in 5730 naturalized alien plants with economic uses. Third, as species that are widely naturalized most likely were also introduced to more regions, we restricted the analysis to plants that have naturalized in at least 10 regions (3001 species) and those that have naturalized in at least 2 regions (7341 species), corresponding to the 75th and 50th percentiles of the distribution of the number of naturalized regions per species, respectively. Moreover, considering the potential imbalance between the number of regions where naturalization has succeeded and those where it has failed, we analyzed varying numbers of regions with naturalization failures to assess the robustness of our findings. Specifically, we randomly selected ten, twenty, and fifty times the number of successfully naturalized regions from those not yet observed to have naturalization, without exceeding the total number of non-native regions[75]. These regions were then combined with the successfully naturalized regions to identify potential recipient regions for further analysis. Using these data subsets, we ran the above GLMMs and calculated the standardized coefficients for each ecological distance metric (Supplementary Fig. 6).

We conducted an additional analysis focusing on the naturalization of species native to mainland regions on islands and in other mainland regions (Supplementary Fig. 7). For this analysis, we restricted the dataset to species with native distributions exclusively or partially in mainland regions, resulting in a subset of 11,299 naturalized alien plants (representing 97.4% of all species). Islands and mainland regions were classified according to the definitions provided in the GloNAF database[57]. We ran GLMMs separately for naturalization in mainland and island regions to assess how naturalization probability relates to ecological distance metrics, and we then calculated the standardized coefficients for each distance variable.

While differences in the standardized model coefficients provide useful insights into the relative importance of each ecological distance, we further complemented this with an additional analysis using a bootstrap approach implemented in the R package glmm.hp v.0.1-8[76,77]. This approach estimates variable importance by partitioning the marginal $R^2$ attributable to fixed effects, incorporating both linear and quadratic terms[76,77]. Both theoretical and delta $R^2$ were calculated (Supplementary Table 3), with theoretical $R^2$ values reported in the main text (Fig. 2b). Given the substantial size of our full dataset (~6.9 million observations), running glmm.hp on the entire dataset was not feasible. Instead, we randomly selected 5000 observations for each bootstrap sample. For each sample, we fitted a GLMM (the model used in Fig. 1), and used glmm.hp to estimate the combined contribution of the linear and quadratic terms for each ecological distance. We repeated this procedure 999 times, and the resulting relative importance distributions were visualized using density plots and boxplots.

The above GLMMs were constructed using the R package lme4 v.1.1-35.1[78], and the standardized coefficients (scaled without centering) for the variables were extracted from the corresponding models. We only used two-sided tests. The predicted relationships between naturalization probability and each predictor variable were visualized using the R package ggeffects v.1.5.0[79]. All statistical analyses were conducted in R v.4.3.3[80].

### Reporting summary
Further information on research design is available in the Nature Portfolio Reporting Summary linked to this article.

## Data availability
The data generated in this study have been deposited in figshare (https://doi.org/10.6084/m9.figshare.28513706)[81]. Source data are provided with this paper.

## Code availability
The code for the analyses and figures is available in figshare (https://doi.org/10.6084/m9.figshare.28513706)[81].

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

## Acknowledgements

This project was supported by the National Natural Science Foundation of China (NSFC32222051) and the National Key R&D Program (2023YFC2604500) to S.P.L. S.Y.F. acknowledges the funding from the China Scholarship Council (grant no. 202106140108). F.E. appreciates funding by the Austrian Science Fund FWF (project no. I 5825-B). P.P. and J.P. were supported by long-term research development project RVO 67985939 (Czech Academy of Sciences). M.W., M.v.K., and A.J.S.D. acknowledge funding by the German Research Foundation DFG (FZT 118, 202548816 via iDiv, and 264740629 to M.v.K.). We thank Dr. Jiangshan Lai, author of the glmm.hp package, for his assistance with the analysis of relative variable importance using the random-subset approach.

## Author contributions

S.Y.F., S.P.L., and M.v.K came up with the original idea. S.Y.F. prepared and analyzed the data and wrote the first manuscript draft with major inputs from T.S.F., S.P.L., and M.v.K. P.W., H.K., W.D., M.W., P.P., J.P., F.E., A.J.S.D., and M.v.K. provided data and contributed to the writing.

## Competing interests

The authors declare no competing interests.

## Additional information

¹Research Center for Global Change and Ecological Forecasting, Zhejiang Tiantong Forest Ecosystem National Observation and Research Station, Zhejiang Zhoushan Island Ecosystem Observation and Research Station, Institute of Eco-Chongming, School of Ecological and Environmental Sciences, East China Normal University, Shanghai, China. ²Ecology, Department of Biology, University of Konstanz, Konstanz, Germany. ³Department of Biology, University of Puerto Rico - Río Piedras, San Juan, Puerto Rico. ⁴Department of Environmental Science, Radboud Institute for Biological and Environmental Sciences, Radboud University, Nijmegen, The Netherlands. ⁵Biodiversity, Macroecology & Biogeography, University of Göttingen, Göttingen, Germany. ⁶Campus-Institut Data Science, University of Göttingen, Göttingen, Germany. ⁷Centre of Biodiversity and Sustainable Land Use, University of Göttingen, Göttingen, Germany. ⁸Department of Evolution, Ecology and Behaviour, Institute of Infection, Veterinary and Ecological Sciences, University of Liverpool, Liverpool, UK. ⁹German Centre for Integrative Biodiversity Research (iDiv) Halle-Jena-Leipzig, Leipzig, Germany. ¹⁰Department of Invasion Ecology, Institute of Botany, Czech Academy of Sciences, Průhonice, Czech Republic. ¹¹Department of Ecology, Faculty of Science, Charles University, Prague, Czech Republic. ¹²Division of Bioinvasions, Global Change & Macroecology, Department of Botany and Biodiversity Research, University of Vienna, Vienna, Austria. ¹³Zhejiang Provincial Key Laboratory of Plant Evolutionary Ecology and Conservation, Taizhou University, Taizhou, China. ¹⁴Zhejiang Key Laboratory for Restoration of Damaged Coastal Ecosystems, School of Life Sciences, Taizhou University, Taizhou, Zhejiang, China. ✉e-mail: spli@des.ecnu.edu.cn

