## [Transparent Peer Review file · Nature Communications]

Ecological similarities and dissimilarities between donor and recipient regions shape global plant naturalizations

Corresponding Author: Dr Shaopeng Li

Version 0:

Reviewer comments:

Reviewer #1

(Remarks to the Author)

This is a potentially important paper that at present, in my opinion, isn't quite living up to its potential. The paper does a masterful job of looking at a wide variety of very different types of factors that might influence patterns of plant naturalization globally. It is unique, to the best of my knowledge, in taking this question on at a global scale with plants. I'm particularly impressed with the rigor they apply to measuring the variables of interest and controlling for things like differences in area of the regions examined. They have managed to integrate in a single study factors as disparate as phylogenetic diversity of regions, climate and human disturbance. I think with a little additional work that this manuscript could be highly influential.

My biggest criticism of the paper is that it doesn't fully develop a measure of the relative importance of the many factors they find to be important in predicting plant naturalizations. This information is buried in Fig. 2, where they show the effect size of the various variables, but that's it. That's the only place in the ms that it's shown and these various effect sizes aren't discussed in any way. They also don't list these values out in a suppl. table, but they should. I suggest that these differences should first be mentioned in the Results section and then interpreted in the Discussion section. Most readers (me included) would benefit from a discussion of whether a large coefficient with the linear vs. the quadratic term matters in some way. More importantly, the paper would benefit from discussing the relative importance of these variables. It looks like the single most important variable is phylogenetic diversity difference. Is this correct? If so, this should be explained and contextualized. Why might this single variable be so important relative to the others? Likewise, temperature difference also looks to be nearly as important, whereas all the other variables have much smaller coefficients. Does this mean that if you know the temperature and phylogenetic differences that you can predict most of the naturalizations? I'm really not sure, but this is the sort of thing that should be explained. At present, it is entirely left to the reader to wonder about these differences and how to interpret them. It would also be really useful to relate the relative importance of these variables to other studies that have tried to predict invasions to single regions (e.g., all the work done to try to predict what species pose an invasion risk in Australia). Are they finding the same variables are important and how should these similarities or differences be interpreted? Developing this aspect of the paper is the main thing that I believe is needed.

Personally, I think it would be ideal to list all the results in the Results section. At present there is some new results (e.g., differences between islands and mainlands) presented for the first time in the discussion section.

The Discussion section needs some attention. The first paragraph feels largely like a restating of the Results section. Instead, making the main points or a major finding would be more ideal. I'm more concerned, however, about the places where correlations seem to be confounded with possible explanations that are listed as if they are the only possibility. A prime example is the discussion around the importance of climate matching. The ms makes it sound like climate preadaptation is the explanation for this correlation. It certainly might be the explanation, but another reasonable possibility is that people tend to introduce plants between regions with similar climates because they think they'll do well there. I don't know how important those two possible explanations might be, but the paper should be clear that more than one explanation is possible. Likewise, there are places in the Discussion where difference in patterns of trade could be important in interpreting results, but this isn't mentioned. Another example is with human disturbance. They explain that these areas might have changed environments that promote invasion. This could totally be true, but an alternative is that highly disturbed regions are also regions with lots of people and lots of trade and lots of introduced species, so that this is what matters and not the disturbance per se. More generally, I'd like to see the Discussion section do a more thorough job of explaining some of the possibilities and be more careful not to cherry pick particular explanations and make them sound like the only ones

that could explain these patterns.

I'd like to see the abstract list out results with more specificity.

(Remarks on code availability)

Reviewer #2

(Remarks to the Author)

This article was written by a very experienced team that is not new to working with a huge database on exotic plants from around the world (11,604 species). The text is very well written and the approach proposed is coherent. It is difficult to understand all the ins and outs of the statistical methodology, which is very sophisticated, but I have few fears regarding its reliability, considering that this is not the first experience of this team in this area. I have a few more reservations about the human modification index (HMI) used to describe the degree of human-caused landscape modification, which seems a bit crude to me, but I see little other avenue to explore to include in the model the impact of the transformation of natural habitats. I am still quite surprised, particularly in relation to this index, by the relatively small difference in probability that is illustrated in each element of Figure 1 (the shaded areas representing the 95% confidence intervals). This small difference is surprising given the diversity of plants and regions involved. Nevertheless, it gives an excellent idea of the strength of the model, which is remarkable.

The question that arises with this article is: what new information do we learn from reading it? In essence, the authors argue (rightly) that the results reveal that naturalization of an alien plant in a recipient region is facilitated by similarity to the donor regions in temperature and floristic composition of the native floras, and that species are more likely to successfully naturalize in recipient regions with higher levels of human disturbance and lower native flora diversity than in their native range. No one should be surprised by this conclusion; it was expected, whatever the authors say in the introduction. Does this mean that the article is irrelevant? No, it is, in the sense that it supports with a solid database and a robust methodology what researchers have suspected for decades, but on a regional basis – here, the authors offer a global perspective. Where it falls a little short is in the explanations provided in the discussion, which by necessity are rather vague and quite speculative (ex.: species in their native ranges do not fully occupy their fundamental climatic niches due to biotic interactions, particularly in favorable climates; our finding supports the potential importance of pre-adaptation to climatic and biotic conditions in facilitating global plant naturalization; human-modified environments often offer additional resources and opportunities for establishment). I am well aware that in a global analysis it is difficult to go into great detail, but the interpretation of the results is not the strongest point of this manuscript. In any case, this comment does not detract from the intrinsic quality of the work, which constitutes a solid contribution to the development of theoretical knowledge on biological invasions.

(Remarks on code availability)

Version 1:

Reviewer comments:

Reviewer #1

(Remarks to the Author)

I am very pleased with the revisions made to the manuscript. It was a strong paper to begin with, but the revised manuscript is now greatly strengthened. I suspect this will be a highly influential paper.

I have no suggestions for additional work or revisions. I will mention one paper I authored (long ago) that does seem relevant to this work that I want to bring to the author's attention. Specifically, Sax, D.F., 2001, *Journal of Biogeography* 28: 139-150. The paper is mostly focused on latitudinal gradients of non-native species, but comparisons are also made between the latitudinal boundaries of species native and naturalized ranges. The work finds with a global analysis of birds that when birds naturalize on continents that they almost never occupy latitudes that are lower than those found in the native range. In contrast, when birds naturalize on islands they are often able to occupy latitudes lower than those occupied in the native range. The work suggests that when biotic pressures are relaxed (in lower diversity environments) that additional types of climatic conditions become suitable. This is a result that is highly consistent with what you've found in your work for plants.

Best regards,
Dov Sax

(Remarks on code availability)

Reviewer #2

(Remarks to the Author)

No additional corrections required.

(Remarks on code availability)

Responses to the comments

Responses to the Reviewer 1's comments

This is a potentially important paper that at present, in my opinion, isn't quite living up to its potential. The paper does a masterful job of looking at a wide variety of very different types of factors that might influence patterns of plant naturalization globally. It is unique, to the best of my knowledge, in taking this question on at a global scale with plants. I'm particularly impressed with the rigor they apply to measuring the variables of interest and controlling for things like differences in area of the regions examined. They have managed to integrate in a single study factors as disparate as phylogenetic diversity of regions, climate and human disturbance. I think with a little additional work that this manuscript could be highly influential.

Response:

Thank you for your positive comments and constructive suggestions. In response to your valuable feedback provided below, we have carefully revised our manuscript and incorporated additional analyses.

Comments:

My biggest criticism of the paper is that it doesn't fully develop a measure of the relative importance of the many factors they find to be important in predicting plant naturalizations. This information is buried in Fig. 2, where they show the effect size of the various variables, but that's it. That's the only place in the ms that it's shown and these various effect sizes aren't discussed in any way. They also don't list these values out in a suppl. table, but they should. I suggest that these differences should first be mentioned in the Results section and then interpreted in the Discussion section. Most readers (me included) would benefit from a discussion of whether a large coefficient with the linear vs. the quadratic term matters in some way. More importantly, the paper would benefit from discussing the relative importance of these variables. It looks like the single most important variable is phylogenetic diversity difference. Is this correct? If so, this should be explained and contextualized. Why might this single variable be so important relative to the others? Likewise, temperature difference also looks to be nearly as important, whereas all the other variables have much smaller coefficients. Does this mean that if you know the temperature and phylogenetic differences that you can predict most of the naturalizations? I'm really not sure, but this is the sort of thing that should be explained. At present, it is entirely left to the reader to wonder about these differences and how to interpret them. It would also be really useful to relate the relative importance of these variables to other studies that have tried to predict invasions to single

regions (e.g., all the work done to try to predict what species pose an invasion risk in Australia). Are they finding the same variables are important and how should these similarities or differences be interpreted? Developing this aspect of the paper is the main thing that I believe is needed.

Response:

Thank you for this thoughtful and constructive comment! We fully agree that clearly presenting and interpreting the relative importance of predictors is essential for our study. In response, we have made substantial revisions to the main text and the supplementary information to address this point.

First, we conducted a new bootstrap-based analysis to assess the relative importance of each ecological distance metric (see Methods, lines 546–557). Specifically, we fitted 999 models to randomly selected subsets of 5,000 observations and calculated the proportion of variance explained (R^2) by each predictor in each model. We then averaged these values across replicates to estimate the mean contribution of each variable. The results are presented in a new panel (Figure 2b) and detailed in the Results (lines 165–184), complementing the standardized regression coefficients shown in Figure 2a. This new analysis confirmed temperature similarity and native flora phylogenetic diversity distance as the most consistently influential predictors of naturalization probability.

Second, to further enhance transparency and interpretability, we added two supplementary tables. Supplementary Table 2 lists the standardized coefficients for the linear and quadratic terms of each variable, as shown in Figure 2a. Supplementary Table 3 presents the mean relative importance values obtained from the bootstrap procedure, corresponding to Figure 2b. Together with Figure 2, these tables offer a detailed numerical basis for interpreting the magnitude and shape of each variable's effect on naturalization probability.

Third, we substantially expanded the Discussion to better highlight the relative importance of different ecological variables (lines 209–220, 248–256, 299–308, 318–323). We clarified that temperature similarity and native flora phylogenetic diversity distance are the most important predictors of naturalization probability and discussed the potential mechanisms behind each: the importance of temperature similarity likely reflects climatic pre-adaptation and human-mediated introduction preferences (lines 211–220), while the prominent role of native flora phylogenetic diversity distance may reflect the influence of the evolutionary history of native floras on naturalization outcomes (lines 313–316, 318–321). We also noted the relevance of these findings for future predictive research, emphasizing that the study provides an important foundation for understanding and predicting global naturalization patterns (lines 257–259, 308–311, 352–355).

Finally, in the Discussion, we now better contextualize our findings by comparing them with results from relevant regional studies (lines 214–217, 250–256, 276–280, 299–308, 321–323). We clarified that the high importance of temperature similarity is consistent with previous studies that identify climate matching as a key predictor of naturalization success (lines 209–213). In contrast, the role native flora phylogenetic diversity distance has received relatively little attention, although several recent studies have begun to acknowledge its relevance (lines 316–323). However, in contrast to some regional studies, our analysis did not identify floristic similarity between native floras or human modification distance as dominant predictors at the global scale (lines 248–252, 299–302). Nevertheless, its weak but statistically significant effect suggests that it may still play a non-negligible role. We further discussed possible reasons for observed differences between our results and those of previous studies, including differences in the invasion stages, the spatial extent, and the specific predictors considered (lines 252–256, 302–308).

In summary, thank you again for encouraging us to strengthen this central aspect, which we believe significantly enhance the depth and clarity of our manuscript.

Personally, I think it would be ideal to list all the results in the Results section. At present there is some new results (e.g., differences between islands and mainlands) presented for the first time in the discussion section.

Response:

As suggested, we have revised the manuscript to ensure that all results, including the differences in naturalization patterns between island and mainland regions, are now clearly presented in the Results section (lines 192–194). This ensures that the Discussion focuses exclusively on the interpretation and discussion of the results, without introducing new results.

The Discussion section needs some attention. The first paragraph feels largely like a restating of the Results section. Instead, making the main points or a major finding would be more ideal.

Response:

As suggested, we have revised the opening paragraph of the Discussion to avoid repeating detailed results and instead provide a concise synthesis of the main findings. Specifically, we now emphasize that “Our results reveal that similarity in climate and differences in native plant diversity between donor and recipient regions are the most important factors shaping

global patterns of plant naturalization. Alien plants were more likely to naturalize in regions with temperature conditions similar to their donor regions, while differences in other environmental factors, such as moderately wetter and lower native flora diversity in recipient regions, also facilitated naturalization success.” (lines 199–204).

I’m more concerned, however, about the places where correlations seem to be confounded with possible explanations that are listed as if they are the only possibility. A prime example is the discussion around the importance of climate matching. The ms makes it sound like climate preadaptation is the explanation for this correlation. It certainly might be the explanation, but another reasonable possibility is that people tend to introduce plants between regions with similar climates because they think they’ll do well there. I don’t know how important those two possible explanations might be, but the paper should be clear that more than one explanation is possible.

Response:

Thank you for raising this important point. We fully agree that correlations should not be interpreted as evidence for a single causal mechanism, and that alternative explanations should be acknowledged. In the revised Discussion (lines 213–220), we have added a clarification noting that both pre-adaptation and human preference for introducing plants between climatically similar regions are plausible and potentially complementary explanations for the observed patterns. We carefully revised the relevant sentences and hope this revision offers a more balanced interpretation of the observed patterns (lines 207–220)

Likewise, there are places in the Discussion where difference in patterns of trade could be important in interpreting results, but this isn’t mentioned. Another example is with human disturbance. They explain that these areas might have changed environments that promote invasion. This could totally be true, but an alternative is that highly disturbed regions are also regions with lots of people and lots of trade and lots of introduced species, so that this is what matters and not the disturbance per se. More generally, I’d like to see the Discussion section do a more thorough job of explaining some of the possibilities and be more careful not to cherry pick particular explanations and make them sound like the only ones that could explain these patterns.

Response:

We appreciate this thoughtful suggestion and fully agree that multiple alternative explanations may underlie the observed patterns. As recommended, we have revised the Discussion to acknowledge that the observed associations with human modification may reflect not only

environmental changes, but also broader socio-economic factors such as population density, trade volume, and transport connectivity (lines 286–291). We have also highlighted the potential influence of trade asymmetry, such as biased flows of goods between regions, on the directional pressure of human-mediated plant introductions (lines 291–294). These additions aim to provide a more comprehensive and balanced interpretation of our findings.

I'd like to see the abstract list out results with more specificity.

Response:

As suggested, we revised the abstract to make it more specificity. In particular, we clarified that “Among all predictors, climate similarity and difference in native flora diversity emerged as the strongest predictors of naturalization success.” (lines 50–52).

Responses to the Reviewer 2's comments

This article was written by a very experienced team that is not new to working with a huge database on exotic plants from around the world (11,604 species). The text is very well written and the approach proposed is coherent. It is difficult to understand all the ins and outs of the statistical methodology, which is very sophisticated, but I have few fears regarding its reliability, considering that this is not the first experience of this team in this area. I have a few more reservations about the human modification index (HMI) used to describe the degree of human-caused landscape modification, which seems a bit crude to me, but I see little other avenue to explore to include in the model the impact of the transformation of natural habitats.

Response:

Thank you for your thoughtful and encouraging feedback. We agree that the Human Modification Index (HMI), while a widely used and globally available measure, provides a relatively coarse representation of human-driven environmental change. We chose to include HMI in our analysis because it integrates multiple components of anthropogenic disturbance, enabling consistent comparisons across regions at a global scale. In the revised Discussion (lines 306–308), we explicitly note that the HMI may not fully capture the specific dimensions of human influence most relevant to naturalization processes, such as trade volume, transportation connectivity, or propagule pressure. We also suggest that future research could benefit from incorporating finer-scale or more targeted socio-economic and ecological indicators, where available, to better capture the role of human activity in shaping global naturalization patterns (lines 308–311).

I am still quite surprised, particularly in relation to this index, by the relatively small difference in probability that is illustrated in each element of Figure 1 (the shaded areas representing the 95% confidence intervals). This small difference is surprising given the diversity of plants and regions involved. Nevertheless, it gives an excellent idea of the strength of the model, which is remarkable.

Response:

We sincerely appreciate your thoughtful observation. We'd like to clarify that the relatively small differences in predicted probabilities shown in Figure 1 are a direct reflection of the structure of our data and the modeling framework. Specifically, the response variable represents the average probability of naturalization for each species across all potential recipient regions. Since most species are naturalized in only a limited number of regions, these average probabilities are generally low. Moreover, our binomial models incorporated

strong random effects for both species and regions to account for unobserved heterogeneity, which also contribute to more conservative and stable predictions. We also performed several sensitivity analyses, including analyses across different sets of potential recipient regions and subsets of naturalized alien plants (Supplementary Fig. 6). All results consistently supported the main conclusions, reinforcing the strength and reliability of the patterns shown in Figure 1.

The question that arises with this article is: what new information do we learn from reading it? In essence, the authors argue (rightly) that the results reveal that naturalization of an alien plant in a recipient region is facilitated by similarity to the donor regions in temperature and floristic composition of the native floras, and that species are more likely to successfully naturalize in recipient regions with higher levels of human disturbance and lower native flora diversity than in their native range. No one should be surprised by this conclusion; it was expected, whatever the authors say in the introduction. Does this mean that the article is irrelevant? No, it is, in the sense that it supports with a solid database and a robust methodology what researchers have suspected for decades, but on a regional basis – here, the authors offer a global perspective.

Response:

Thank you for this thoughtful assessment. We are glad that the reviewer acknowledges the value of our global perspective and robust methodology in reinforcing and extending regional-scale findings. In addition, we believe our study also offers novel insights by introducing a comprehensive analytical framework to quantify the relative importance of multiple ecological distances, including both bidirectional and unidirectional metrics. Our results not only confirm long-held assumptions but also challenge the traditional view that naturalized species are primarily successful in regions with conditions similar to their native ranges. We show that dissimilarity in specific directions and dimensions, such as lower native plant diversity compared to donor regions, can also strongly promote naturalization at the global scale, offering a more comprehensive understanding of the uneven global distribution of naturalized species.

Our findings reveal that plants were more likely to naturalize in regions where climatic conditions and the phylogenetic composition of native floras closely resembled those of their donor regions. However, differences in certain ecological factors—such as lower native plant diversity and higher levels of human activity compared to donor regions—further promoted species naturalization. This challenges the conventional view that naturalized species are primarily successful in regions with

similar conditions to their native ranges, offering a more comprehensive understanding of the uneven global distribution of naturalized species.

Where it falls a little short is in the explanations provided in the discussion, which by necessity are rather vague and quite speculative (ex.: species in their native ranges do not fully occupy their fundamental climatic niches due to biotic interactions, particularly in favorable climates; our finding supports the potential importance of pre-adaptation to climatic and biotic conditions in facilitating global plant naturalization; human-modified environments often offer additional resources and opportunities for establishment). I am well aware that in a global analysis it is difficult to go into great detail, but the interpretation of the results is not the strongest point of this manuscript. In any case, this comment does not detract from the intrinsic quality of the work, which constitutes a solid contribution to the development of theoretical knowledge on biological invasions.

Response:

Thank you for this thoughtful and constructive feedback. As suggested, we have made greater efforts to improve clarity and depth in the revised Discussion. Specifically, we have revised the statements regarding species' climatic niches (lines 225–228) and human-modified environments (lines 286–288), and we have removed the sentence that “our findings support the potential importance of pre-adaptation to climatic and biotic conditions” to avoid overstatement. In addition, we have added more nuanced and balanced interpretations of our key findings. For example, regarding temperature similarity, we consider both the role of climatic pre-adaptation and the potential roles of human-mediated introduction biases (lines 211–220). In interpreting the effect of human modification distance, we have expanded our interpretation to consider not only landscape transformation but also broader socio-economic factors, including population density, trade volume, and transport connectivity, which may drive introduction pressure (lines 286–291). We also briefly consider the role of trade asymmetry as a potential contributing factor (lines 291–294). We also highlight promising future directions for investigating these potential mechanisms in greater detail than is possible in our global scale analyses (lines 246–248, 257–259, 273–276, 308–311, 334–337, 341–343). Together, we hope these revisions could present a more nuanced and well-rounded interpretation of our global findings.

Responses to the comments

Responses to the Reviewer 1's comments

I am very pleased with the revisions made to the manuscript. It was a strong paper to begin with, but the revised manuscript is now greatly strengthened. I suspect this will be a highly influential paper.

I have no suggestions for additional work or revisions. I will mention one paper I authored (long ago) that does seem relevant to this work that I want to bring to the author's attention. Specifically, Sax, D.F., 2001, *Journal of Biogeography* 28: 139-150. The paper is mostly focused on latitudinal gradients of non-native species, but comparisons are also made between the latitudinal boundaries of species native and naturalized ranges. The work finds with a global analysis of birds that when birds naturalize on continents that they almost never occupy latitudes that are lower than those found in the native range. In contrast, when birds naturalize on islands they are often able to occupy latitudes lower than those occupied in the native range. The work suggests that when biotic pressures are relaxed (in lower diversity environments) that additional types of climatic conditions become suitable. This is a result that is highly consistent with what you've found in your work for plants.

Response: Thank you for this positive evaluation of our work. We also greatly appreciate the suggestion of Sax (2001), which is highly relevant to our study. We have now added a citation to this reference and briefly discussed its relevance in the revised manuscript. In particular, we clarified that “Similar patterns have been observed in other taxonomic groups. A global analysis of birds found that naturalized species on islands, where native diversity and ecological resistance are generally lower, are often able to occupy latitudes lower than those in their native ranges. These low-latitude regions are usually characterized by warmer and wetter climates.” (lines 249 – 253).

Responses to the Reviewer 2's comments

No additional corrections required.

Response: Thank you for the positive assessment of our revised manuscript and for confirming that no further corrections are required.